# Specific Heat Capacity and Thermal Conductivity Measurements of PLA-Based 3D-Printed Parts with Milled Carbon Fiber Reinforcement

**DOI:** 10.3390/e24050654

**Published:** 2022-05-06

**Authors:** Ignazio Blanco, Gianluca Cicala, Giuseppe Recca, Claudio Tosto

**Affiliations:** 1Department of Civil Engineering and Architecture and UdR-Catania Consorzio INSTM, University of Catania, Via Santa Sofia 64, 95125 Catania, Italy; iblanco@unict.it (I.B.); gcicala@unict.it (G.C.); 2Institute for Polymers, Composites and Biomaterials, IPCB-CNR, Via Gaifami 18, 95126 Catania, Italy; giuseppe.recca@cnr.it

**Keywords:** heat capacity, thermal conductivity, 3D printing, DSC, laser flash

## Abstract

This research focuses on the thermal characterization of 3D-printed parts obtained via fused filament fabrication (FFF) technology, which uses a poly(lactic acid) (PLA)-based filament filled with milled carbon fibers (MCF) from pyrolysis at different percentages by weight (10, 20, 30 wt%). Differential scanning calorimetry (DSC) and thermal conductivity measurements were used to evaluate the thermal characteristics, morphological features, and heat transport behavior of the printed specimens. The experimental results showed that the addition of MCF to the PLA matrix improved the conductive properties. Scanning electron microscopy (SEM) micrographs were used to obtain further information about the porosity of the systems.

## 1. Introduction

With the introduction of Industry 4.0, additive manufacturing (AM), in which the most widespread process is certainly fused deposition modelling (FDM), has become a prominent player in industrial engineering [1,2,3,4]. FDM filaments containing carbon fibers (CFs) are particularly appealing for applications in automotives, robotics, drones, prosthetics, and orthotics [5,6]. Researchers have focused their attention on bioplastic filaments, particularly PLA, for various structural and biomedical applications due to growing concerns about environmental pollution and a reduction in the availability of petroleum-based plastics [7,8,9]. Reinforcements are also utilized to enhance the PLA’s strength [10]. When compared to petroleum-based acrylonitrile butadiene styrene, polyethylene, and polypropylene, PLA is a more environmentally friendly polymer; thus, its production and use are likely to increase in the next years. As a result, there is a pressing need to better comprehend and improve its characteristics.

The mechanical properties of FDM printed parts made of PLA composite filaments have sparked a lot of attention in the literature [11,12,13,14]. According to some studies, adding expanded graphite nanofillers to PLA-based materials enhanced their tensile and storage modulus, as well as their thermal conductivity [15,16,17]. Unlike the impact of printing parameters on the mechanical properties of FDM printed parts, which have been thoroughly investigated, the parts’ effective thermal conductivity is still little understood [18]. As reported in the literature, the role of thermal characterization [19], and, in particular, of thermal diffusivity [20], is important to revealing the structural characteristics of FDM printed samples characterized by different infill percentages. Patti et al. [20] found an increase of 77% in thermal diffusivity in correspondence to an infill level of 100% compared to 0%. Relating the thermal properties and the various structural parameters of samples obtained by FDM is an important aspect, especially for this technique governed by multiple parameters that affect the quality of the final products. In fact, the development of inherent defects during the production process, such as void formation, surface roughness, and poor bonding between the fiber and matrix, reduces the performance of 3D-printed parts compared to that achieved by conventional manufacturing processes [21,22]. Stepashkin et al. [23] used a unique FDM process to prepare composite systems made of CFs and polyether ether ketone and investigated the corresponding properties in terms of heat conductivity and microscopic pictures. Testing revealed that fiber orientation along the polymer flow caused thermal anisotropy, as well as the existence of cracks and flaws caused by thermal and mechanical stress during filament deposition [23,24,25].

In this study, milled carbon fibers (MCFs) from pyrolysis were utilized to fill a PLA filament for use in FDM, which was then characterized by DSC and thermal diffusivity testing. The influence of varied filler percentages by weight (10, 20, 30 wt%) on the thermal behavior was studied. Final considerations regarding morphological aspects are also reported.

## 2. Materials and Methods

### 2.1. Material and Sample Preparation

The PLA grade 4032D from Natureworks LCC (Blair, NE, USA) was chosen because it is a commonly used polymer for FDM printing. Easycomposites (Stoke-on-Trent, UK) provided MCFs generated by pyrolysis. A Brabender GmbH & Co. (Duisburg, Germany) Compounding KETSE 20/40 twin-screw extruder was used to make the filaments. For this work, four formulations were studied, for which the percentages by weight of pure PLA and MCFs are reported in Table 1.

The PLA pellets were first dried in an industrial oven at a temperature of 50–60 °C to avoid creating air bubbles in the filament and clogging the nozzle during extrusion. The filament was extruded with a diameter of 2.85 ± 0.2 mm. The Ultimaker S5 printer (Figure 1) was used to create the printed samples for thermal diffusivity testing.

### 2.2. C_p_ Measurements

The specific heat capacity (*C_p_*) of the materials was measured by DSC analysis, using a Mettler DSC1 Star System (Mettler Instrument, Greifensee, Swizterland). The calorimeter was calibrated, in enthalpy and temperature, following a well-consolidated procedure [26]. The glass transition temperature (*T*_g_), the cold crystallization temperature (*T*_c_), the enthalpy of cold crystallization (Δ*H*_c_), the melting temperature (*T*_m_), and the enthalpy of melting (Δ*H*_m_) were determined performing three scans for each sample and reported as the averaged value; the deviation remained within ±0.5%. Samples of about 8.0 × 10^–3^ g, held in sealed aluminum crucibles, were heated, with a scanning rate of 10 °C min^−1^ and a flowing nitrogen atmosphere, in the range 25–250 °C.

### 2.3. Thermal Diffusivity

The thermal diffusivity tests were performed on an LFA 467 HT HyperFlash machine (NETZSCH-Gertebau GmbH, Selb, Germany), in accordance with ASTM E1461, on square samples with sides of 12.7 mm and a thickness of 2 mm, covered by graphite. Thermal diffusivity experiments were carried out at temperatures ranging from room temperature to 90 °C, with measurements taking place in isothermal conditions at 70 and 90 °C.

### 2.4. Scanning Electron Microscopy (SEM)

The sample cross section after gold sputtering was examined using an emission scanning electron microscope (SEM EVO by Zeiss, Cambridge, UK) working in high vacuum settings. Using an Emitech K-550 sputter coater (Ashford Kent, UK), all the samples were gold sputtered to a thickness of 20 nm. The micrographs were collected using a 15 kV accelerating voltage and 300× magnification.

## 3. Results

### 3.1. DSC Analysis

The specific heat capacity of the different formulations (Table 1) was computed by DSC.

Figure 2 shows representative DSC thermograms of the investigated samples, whilst the thermal parameters obtained by DSC analysis are summarized in Table 2.

The PLA filament showed cold crystallization at 107.4 °C, followed by a melting peak at 171.4 °C. Unlike the melting, which seems to not be influenced by the MCF content, the cold crystallization temperature of the composites decreased with the inclusion of MCFs, as can be easily seen from Figure 2 and from the data shown in Table 2. From the same Figure 2, it is worth noting a slight increase in the glass transition of the filaments with an increase in MCF content. The *C_p_* values were measured, via DSC, by heating each sample from room temperature up to a temperature just below melting (150 °C), then cooling to room temperature and re-heating up to the same temperature of the first scan. The thermograms obtained were useful for extrapolating, for each sample, the values of *C_p_* in the range of temperatures that goes from shortly after the end of each *T*_g_ to the value of 90 °C—the limit of application interest for these reinforced components.

### 3.2. Thermal Diffusivity Testing

The thermal conductivity of the carbon-fiber-reinforced samples was determined following ASTM E1461, by performing tests on square samples with sides of 12.7 mm and a thickness of 2 mm. Equation (1) was used to calculate thermal conductivity:(1)$$k=\alpha \cdot \rho \cdot C_{p}$$
where *α* is the thermal diffusivity (m^2^ s^−1^), *k* is the thermal conductivity (W m^−1^ K^−1^), *ρ* is the density (kg m^−3^), and *C_p_* is the specific heat capacity (J kg^−1^ K^−1^). To calculate the thermal diffusivity, LFA Proteus analysis software uses Equation (2) to process the time and temperature data collected during the test in adiabatic conditions:(2)$$\alpha =0.1388\frac{s^{2}}{t_{0.5}}$$
where *s* is the thickness of the sample and *t*_0.5_ is the time needed to increase the temperature by 50%.

The standard flash method [27] assumes that the pulse energy is completely absorbed on the specimen’s front face, then migrates through the thickness of the specimen as a thermal wave before reaching the opposite face. In a slightly porous or rough-surfaced material, however, the pulse energy absorption is no longer limited to the front face, but extends over a thin layer into the specimen thickness. The absorption layer can be considered as the material’s mean free path for photons. As a result, the initial temperature distribution within the specimen decays exponentially. The penetration effect and the resulting decaying temperature distribution are taken into account in the porous materials model.

Therefore, before testing, all the samples were coated with graphite to eliminate light reflection errors, as the sample must not be translucent in the visible and near-IR wavelength ranges, and to maximize the samples’ absorption and emission capacity.

The effectiveness of the coating was confirmed by the type of the signal resulting from the tests. In fact, as can be seen in Figure 3 (exemplary signal vs. time curve), in correspondence with the times in the area indicated by the red circle, there were no peaks associated with the penetration of the radiation into the IR detector.

Thermal diffusivity experiments were carried out at temperatures ranging from room temperature to 90 °C, with measurements taking place in isothermal conditions at 70 and 90 °C.

These values were selected because the research group uses these materials for hybrid composites manufacturing (i.e., mold, core materials, tools, etc.). The employed prepreg cures at temperatures less than 100 °C, specifically beginning at 70 °C in the industrial oven. Thus, the temperature range 70–90 °C was investigated.

The area of the analyzed sample was equal to 70%. For each temperature, three shots were performed in order to have a significant average and extrapolate, by approximation using the Cowan model with linear baseline, the law of diffusivity.

The values of *α* and *C_p_* are therefore known from the above tests, while the values of the density *ρ* were obtained by means of an analytical balance with a resolution of 0.1 mg, which exploits the Archimedes principle. The square specimens were weighed in air and then in a known-density auxiliary liquid (i.e., distilled water, Figure 4).

Equation (3) was used to calculate the density of the printed MCF-reinforced sample, *ρ*:(3)$$\rho =\frac{w_{air}}{w_{air}-w_{liq}}\left(\rho_{0}-\rho_{L}\right)+\rho_{L}$$
where *w_air_* is the weight of the sample in air, *w_liq_* is the weight of the sample in the auxiliary liquid, *ρ_L_* is the density of the auxiliary liquid, and *ρ*_0_ is the density of air.

Once the values of *α*, *ρ*, and *C_p_* have been calculated, the thermal conductivity values *k* are obtained by applying Equation (1). The thermal conductivity values of the three MCF-reinforced samples at the investigated temperatures, 70 and 90 °C, are reported in Table 3. The thermal conductivity curves for the three MCF-reinforced samples are reported in Figure 5 and Figure 6.

Observing the plot in Figure 5, deriving from the Cowan model, the thermal conductivity remains constant for PLA with 10 wt% MCFs; a slight increase at high temperatures is appreciable for the grade filled with 30 wt% MCFs; and a greater increase is observed for the grade of 20 wt% MCF content. The data in Table 3 were used to plot the thermal conductivity as the MCF content changes (Figure 6). As expected, the thermal conductivity increases as the MCF content increases.

### 3.3. Morphological Analysis

As reported above, the non-deterioration of the thermal properties of the samples is due to good interlayer bonding during the printing. This result is, first of all, observable from the good surface finish of the printed samples and from the optical images of their cross sections (Figure 7). In fact, compared to unfilled filaments, it is known that fillers, especially if conductive, improve the surface quality and bonding between the layers due to the more favorable conditions for adhesion and diffusion [28,29,30].

The results obtained allow us to speculate that the increase in the thermal diffusivity value is associated with the increase in the MCF content in the 3D-printed composite. To highlight the differences in the distribution of the MCF content in the PLA matrix, sections of the three reinforced samples were observed via SEM. The SEM images are presented in Figure 8.

MCFs spread randomly in the polymer matrix, as shown in Figure 8. A dense conductivity network was generated as the MCF content increased and the fibers touched each other, resulting in the formation of conductive paths.

## 4. Discussion

As expected, the addition of MCFs increased the thermal diffusivity and, as a result, the thermal conductivity, which is beneficial to overall thermal exchange.

Considering the literature data, CFs from polyacrylonitrile have thermal conductivity ranging from 0.43 W m^−1^ K^−1^ [31] to 0.43–0.80 W m^−1^ K^−1^ [32]. Meanwhile, for the PLA used in this study, the thermal conductivity is 0.16 W m^−1^ K^−1^ [33,34,35,36]. Thus, given these considerations, and according to the rule of mixtures, the thermal diffusion in MCF-reinforced samples realized with the extruded filaments (Table 1) should correspond to values of 0.19, 0.21, and 0.24 W m^−1^ K^−1^, respectively, for samples of PLA reinforced with 10, 20, and 30 wt% MCFs.

These values are in agreement with those reported in the literature for other PLA CF-reinforced filaments. Elkholy et al. [18] found for the filament called Robo3D, a CF-reinforced PLA filament with an unknown percentage of fibers, a thermal conductivity value of 0.20 W m^−1^ K^−1^. For another filament called Protopasta—a PLA filled with a maximum of 14.25 wt% short carbon fibers—the same authors found a thermal conductivity value of 0.25 W m^−1^ K^−1^.

In particular, the thermal conductivity values of PLA reinforced with 10 and 20 wt% MCFs are in agreement with the theoretical and literature data. On the other hand, although an increase was observed for the PLA samples with 30 wt% MCFs, there was a small amount of deterioration compared to what was expected.

The greater dependence of thermal conductivity on temperature in the PLA-MCF20 formulation (Figure 5 and Figure 6) can be explained by the better distribution of the reinforcing fibers, together with improved adhesion with the matrix. In fact, as also reported in [34,37], the presence of the conductive reinforcement creates a useful network to increase the thermal and electrical conductivity. The average carbon-free path and the conductive network created by adding conductive filler greatly improved the transport path required for high thermal conductivity.

This network was less pronounced in the formulation with 10 wt% MCFs. Furthermore, the temperature, due to the thermal expansion increasing with the temperature [38], can encourage the formation of wider paths. In fact, the linear thermal expansion coefficient of PLA increases from 7.6 × 10^−5^ to 3.8 × 10^−4^ μm °C^−1^ as the temperature increases from 40 to 80 °C [39]. For these reasons, an appreciable increase in thermal conductivity was noted in the formulation with 20 wt% MCFs. Instead, due to the saturation of the fiber matrix, this increase was less marked in the formulation with MCF reinforcement at 30 wt%.

It is possible that the existence of defects and porosity in the highly filled samples contributed to a small reduction in heat transmission within the material.

In fact, FDM printed parts are inherently anisotropic, even for 100% infill, due to internal pores and air voids [40,41]. As a result, these imperfections limit the contact between the layers and often result in undesirable degradation of the thermal properties.

In the SEM images (Figure 8), it is also possible to observe the porosity within the filaments, mainly observed in the higher grade of reinforcement (PLA_MCF30). This finding was probably due to limited increases in thermal conductivity for this formulation.

In fact, this porosity can be attributed to a lack of interfacial bonding strength between the MCFs and the PLA matrix, which could result in a reduction in the mechanical, thermal, and electrical performance of FDM printed parts [42]. To minimize small intra-filament pores, fiber surface changes with appropriate physical and chemical treatments are beneficial.

It is important to note that the thermal conductivity values are far lower than those obtained by using other types of fillers. For example, Lebedev et al. [43] found that adding graphite to a PLA matrix comparable to the one used in this study increased the thermal conductivity from 0.193 (for PLA) to 2.73 W m^−1^ K^−1^ (for PLA with 30 wt% graphite). Furthermore, by adding 1.0 wt% carbon nanotubes into this PLA–graphite composite, the authors increased the thermal conductivity by approximately 40%, to 3.8 W m^−1^ K^−1^.

Lin et al. [36] found that incorporating graphite nanoplatelets (GNPs) functionalized with tannic acid increased the thermal conductivity of PLA up to 0.77 W∙m^−1^∙K^−1^ (for a 30 wt% content of GNPs).

Mosanenzadeh et al. [35] reached values of 2.77 W m^−1^ K^−1^ by adding 33.3 vol% of hexagonal boron nitride (hBN) or 16.65 vol% of GNPs and hBN to the PLA matrix.

On the other hand, the mechanical properties reached by the samples made with the examined formulations [44] are noteworthy. Indeed, while in Lin et al. [36], the PLA with 8 wt% content of GNPs exhibited a Young’s modulus of 1.46 GPa and tensile strength of 44.1 MPa, for the formulation investigated in this study, PLA with 30 wt% content of MCFs reached a Young’s modulus of 8.56 GPa and tensile strength of 64.59 MPa.

## 5. Conclusions

The effect of MCFs on the thermal characteristics of FDM printed parts was investigated in this study. The *C_p_* value of each of the examined formulations was determined via DSC analyses. These formulations were used to 3D print samples for thermal diffusivity testing. The thermal conductivity was calculated using the *C_p_* values and bulk density data. MCFs increased the thermal diffusivity and, as a result, the thermal conductivity, which is beneficial for thermal performance exchange. The thermal performance was useful in determining whether the 3D-printed parts had any porosity. This approach can be applied and extended to other 3D-printed composite materials, such as the recent hybrid polymer/metal filaments, for which an improvement in printing performance can greatly benefit the debinding and sintering processes to obtain fully dense metal parts.

## Figures and Tables

**Figure 1 entropy-24-00654-f001:**
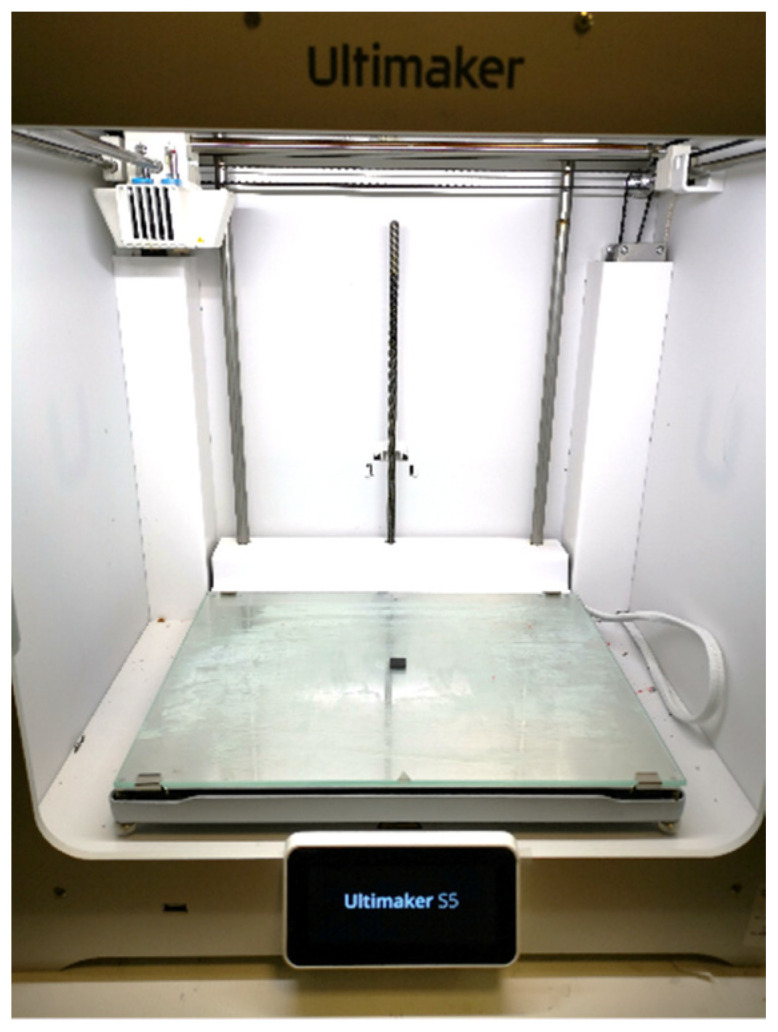
Printing on an FDM desktop machine (Ultimaker S5).

**Figure 2 entropy-24-00654-f002:**
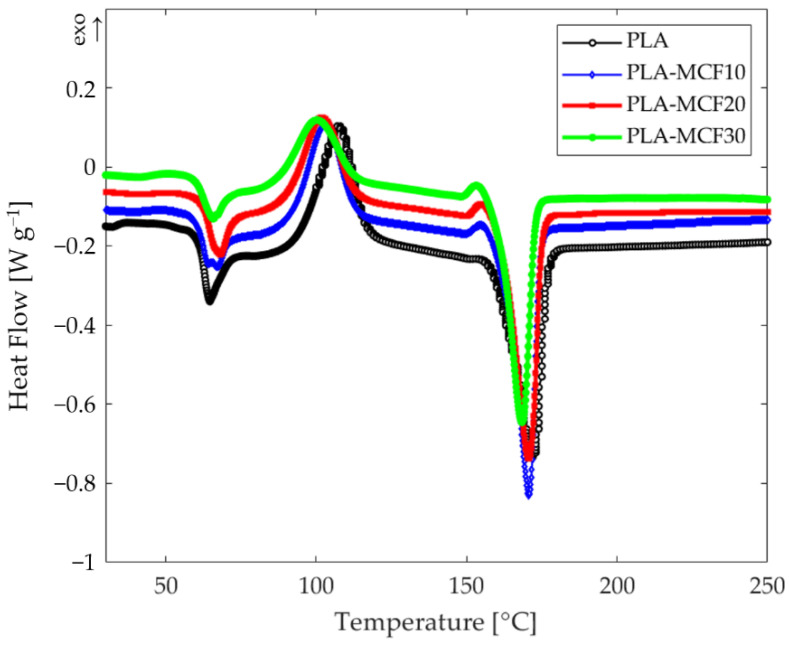
DSC curves for PLA (black) and MCF-reinforced grades at 10 wt% (blue), 20 wt% (red), and 30 wt% (green) MCF content.

**Figure 3 entropy-24-00654-f003:**
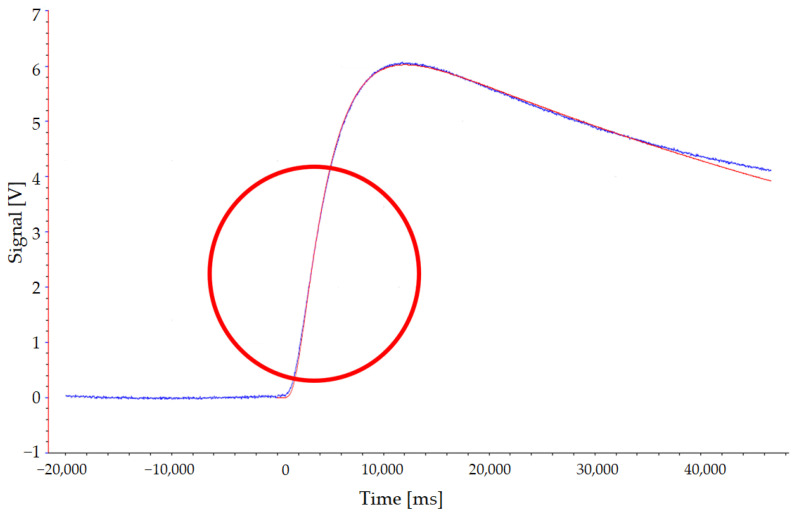
Exemplary curve signal vs. time for model fit (red) and detector (blue).

**Figure 4 entropy-24-00654-f004:**
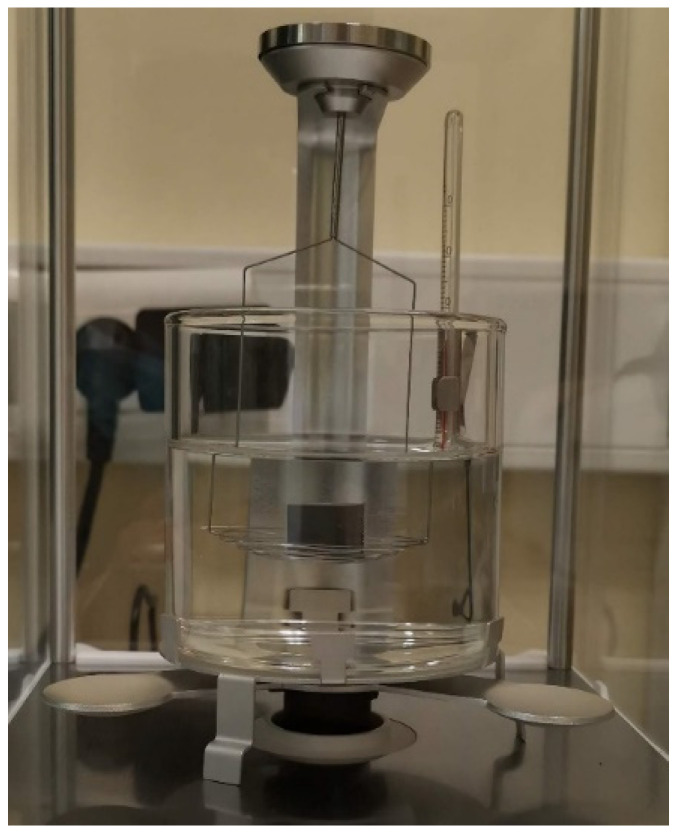
Balance for density determination.

**Figure 5 entropy-24-00654-f005:**
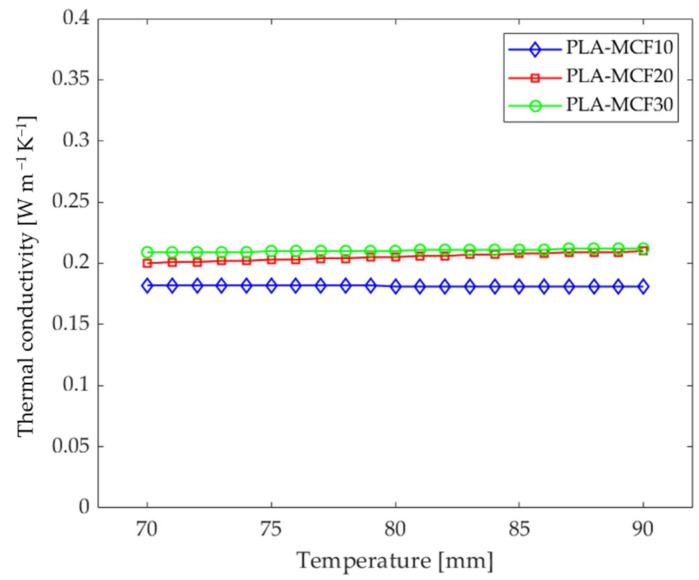
Thermal conductivity vs. temperature curves for the three MCF-reinforced samples.

**Figure 6 entropy-24-00654-f006:**
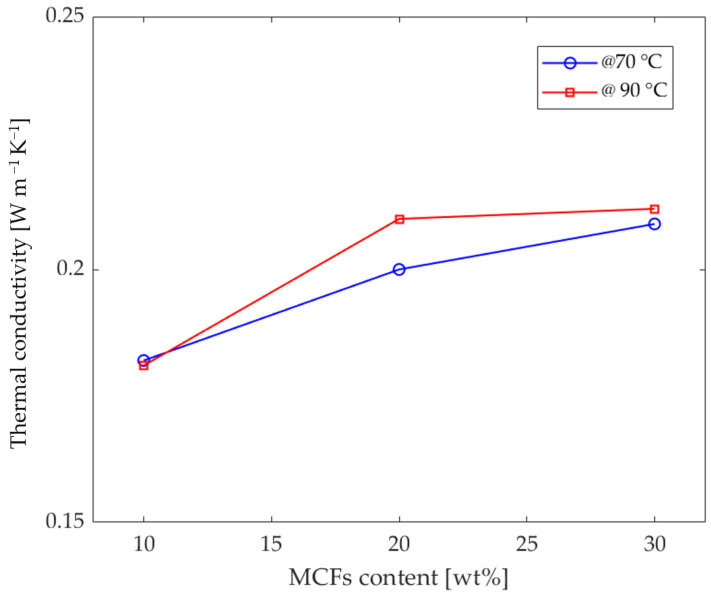
Thermal conductivity vs. MCF content curves for the three MCF-reinforced samples tested at 70 °C (blue) and 90 °C (red).

**Figure 7 entropy-24-00654-f007:**
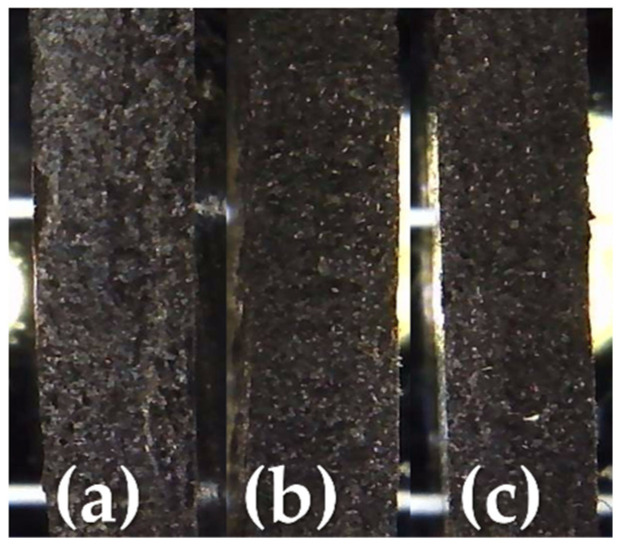
Optical images of the MCF-reinforced samples: (**a**) PLA_MCF10, (**b**) PLA_MCF20, (**c**) PLA_MCF30.

**Figure 8 entropy-24-00654-f008:**
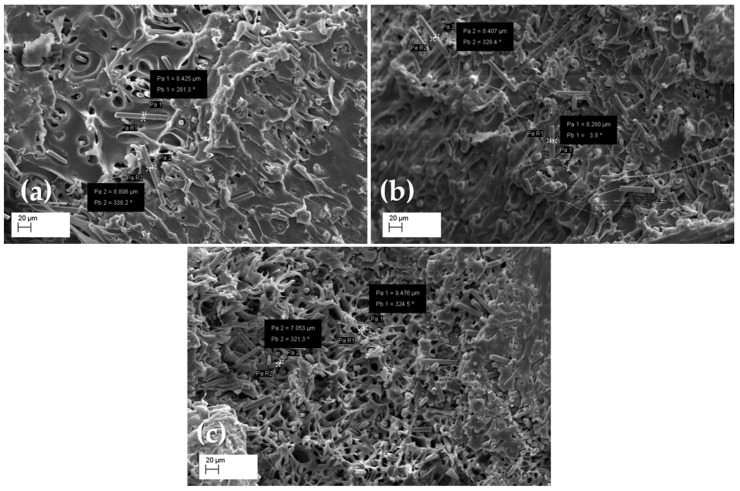
SEM images of the MCF-reinforced samples at 300× magnification for: (**a**) PLA_MCF10, (**b**) PLA_MCF20, (**c**) PLA_MCF30.

**Table 1 entropy-24-00654-t001:** Formulations of the extruded PLA-based filaments.

ID Sample	PLA Content [wt%]	MCF Content [wt%]
PLA	100	0
PLA_MCF10	90	10
PLA_MCF20	80	20
PLA_MCF30	70	30

**Table 2 entropy-24-00654-t002:** Glass transition temperature (*T*_g_), cold crystallization temperature (*T*_c_), enthalpy of cold crystallization (Δ*H*_c_), melting temperature (*T*_m_) and enthalpy of melting (Δ*H*_m_) of the printed samples.

ID Sample	Scan	*T*_g_ [°C]	*T*_m_ [°C]	Δ*H*_m_ [J/g]	*T*_c_ [°C]	Δ*H*_c_ [Jg^−1^]
PLA	I	62. 8	171.4	31.09	107.4	26.81
PLA_MCF10	I	62.1	170.4	32.04	103.0	22.57
PLA_MCF20	I	64.6	170.4	32.40	102.0	20.85
PLA_MCF30	I	64.7	169.5	27.87	101.6	16.50

**Table 3 entropy-24-00654-t003:** Thermal conductivity values of the three MCF-reinforced samples.

	Thermal Conductivity [W m^−1^ K^−1^]	
ID Sample	@ 70 °C	@ 90 °C
PLA_MCF10	0.181 ± 0.001	0.182 ± 0.001
PLA_MCF20	0.200 ± 0.001	0.210 ± 0.001
PLA_MCF30	0.209 ± 0.001	0.212 ± 0.001

## Data Availability

Not applicable.

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
