# Peer review of "Specific Heat Capacity and Thermal Conductivity Measurements of PLA-Based 3D-Printed Parts with Milled Carbon Fiber Reinforcement"

_entropy, 2022, doi:10.3390/e24050654_

Round 1

Author Response

Reviewer 1

 The manuscript entitled ‘Specific heat capacity and thermal conductivity measurements of PLA-based 3D-printed parts with milled carbon fibers reinforcing’ emphasizes on the thermal characterization of the 3D printed parts consisting of varied amounts of milled carbon fibers in poly(lactic acid). The manuscript requires some modification before it is published. My comments on the manuscripts are as follows:

1) MCF has certainly larger thermal conductivity than PLA, but why the increase of the MCF amount from 20 to 30% mixed in PLA enhances thermal conductivity lesser than the increase of MCF from 10 to 20% (Fig. 5)? Couldn’t have the inconsistency of the fabrication process or other factors caused the PLA-MCF 30% to have higher amount of porosity and defects? How is the overall result meaningful?

The reviewer is right, in fact in the manuscript the authors report that " the thermal conductivity remains constant for PLA with 10wt% of MCFs, a slight increase at high temperatures is appreciable for the grade filled with 30 wt% of MCFs, while a greater increase is observed for the grade of 20wt% MCFs content […] It is possible that the existence of defects and porosity in the highly filled samples contributed to a little reduction in heat transmission within the material.". Despite this, the less marked increase in the formulation at 30wt% of MCF is rewarded by the mechanical properties achieved, that are clearly superior to the other two investigated MCF-reinforced formulations and currently not reachable by similar filaments on the market, as reported by the same authors in a paper in press.

Furthermore, in addition to the manufacturing aspect, since Cp comes into play in the thermal conductivity formula, the latter assumes values that are closer to each other for formulations with reinforcement at 20 and 30 wt% of MCF, resulting in conductivity curves with a smaller gap.

2) No increase in the thermal conductivity of PLA-MCF10 with temperature and a slight increase of K with T for PLA-MCF20 and PLA-MCF30 should be explained.

In agreement with the reviewer's suggestions, the following explanation has been added in the revised manuscript.

“The greater dependence of thermal conductivity on temperature in the PLA-MCF20 formulation can be explained by the better distribution of the reinforcing fibers, together with an improved adhesion with the matrix. In fact, as also reported in [33, 36], the presence of the conductive reinforcement creates a useful network to increase the properties of thermal and electrical conductivity. The average carbon-free path and the conductive network created by adding conductive filler greatly improved the transport path required for high thermal conductivity.

This network is less pronounced in the formulation with 10wt% of MCF. Furthermore, the temperature, due to the thermal expansion increasing with the temperature [37] can encourage the formation of wider paths. In fact, the linear thermal expansion coefficient of PLA increases from 7.6∙10-5 to  3.8 ∙10-4 μm∙°C-1 as the temperature increases from 40 to 80 °C [38]. For these reasons, an appreciable increase in thermal conductivity is noted in the formulation with 20wt% of MCF. Instead, due to the saturation of the fiber matrix, this increase is less marked in the formulation with MCF reinforcement at 30wt%”.

3) Paragraphs of Introduction and Discussion are often broken unnecessarily into short paragraphs. They should be rewritten. Full stop is missing in some sentences.

The changes have been implemented and reported in red in the revised manuscript.

Reviewer 2 Report

  1. The abbreviation should be defined upon first mention in abstract as well as main body and can be use directly after it is defined. And, the words mentioned only once do not have to be abbreviated.
  2. Please unify the information description of all materials or equipment. And please provide as much information as possible on all relevant materials or equipment.
  3. “The glass transition temperature (Tg), the temperature (Tc) and enthalpy (ΔHc) of cold crystallization, the onset temperature (Tm) and enthalpy (ΔHm) of melting” This sentence may be ambiguous.
  4. Please unified the use of punctuation marks, such as “°C min−1“.
  5. Please unified the words, such as “neat PLA", "PLA”

Materials and methods:

  1. It is recommended to add a subtitle to this part and correspond to the results section.
  2. It is recommended to expand this part, and supplement the experimental details and instrument parameters.
  3. Please modified the logic of the third paragraph.
  4. Do “The thermal conductivity tests” and “Thermal diffusivity experiments” refer to the same thing?

Result:

  1. What does scan in Table 2 refer to?
  2. What does the red circle part represent? Why this curve was chosen?
  3. Please add more description of Figure 2.
  4. Please revise the logic of the writing order of the materials and methods, results, and discussion parts, referring to other papers in this journal. May some sentences in results need to be put in another part.

Discussion:

Please modify this part referring the comment of result.

Conclusion

Please simplify this part.

Author Response

The abbreviation should be defined upon first mention in abstract as well as main body and can be use directly after it is defined. And, the words mentioned only once do not have to be abbreviated.

Thanks for reporting, the abbreviation have been treated following the reviewer’s suggestion;

Please unify the information description of all materials or equipment. And please provide as much information as possible on all relevant materials or equipment.

All the information are reported, following the Journal template, in the section “Materials and Methods”. The missing information (such as oven model) have been added.

“The glass transition temperature (Tg), the temperature (Tc) and enthalpy (ΔHc) of cold crystallization, the onset temperature (Tm) and enthalpy (ΔHm) of melting” This sentence may be ambiguous.

The reviewer is right the sentence has been modified as follow: “The glass transition temperature (Tg), the cold crystallization temperature (Tc), the enthalpy of cold crystallization (ΔHc), the melting temperature (Tm) and the enthalpy of melting (ΔHm)”

Please unified the use of punctuation marks, such as “°C min−1“.

The correction has been carried out in the text.

Please unified the words, such as “neat PLA", "PLA”.

The text has been modified following the reviewer’s suggestion.

 Materials and methods:

It is recommended to add a subtitle to this part and correspond to the results section.

The text has been modified following the reviewer’s suggestion.

It is recommended to expand this part, and supplement the experimental details and instrument parameters.

All the information are reported, following the Journal template, in the section “Materials and Methods”. The missing information (such as oven model) have been added.

Please modified the logic of the third paragraph.

We are sorry, but we followed the Journal format as well as the logic of our study.

Do “The thermal conductivity tests” and “Thermal diffusivity experiments” refer to the same thing?

Yes, they refer to the same thing. To avoid confusion to readers, the term "Thermal diffusivity" has been kept as the "Thermal conductivity" is a value calculated later.

Result:

What does scan in Table 2 refer to?

In table 2 we reported the thermal parameters measured by DSC analysis, namely The glass transition temperature (Tg), the cold crystallization temperature (Tc), the enthalpy of cold crystallization (ΔHc), the melting temperature (Tm) and the enthalpy of melting (ΔHm). The table caption has been modified to meet the reviewer’s suggestion.

What does the red circle part represent? Why this curve was chosen?

The reviewer is right, changes have been implemented to clarify the importance of the figure in the method for calculating thermal diffusivity.

The curve is exemplary, therefore the caption has been changed in “Exemplary curve signal vs time for model fit (red) and detector (blue)”. In addition, the following part has been included in the revised manuscript.

“The standard flash method [27] assumes that the pulse energy is completely absorbed on the specimen's front face, then migrates through the thickness of the specimen as a thermal wave before reaching the opposite face. In a slightly porous or rough-surfaced material, however, the pulse energy absorption is no longer limited to the front face, but extends over a thin layer into the specimen thickness. The absorption layer can be considered of as the material's mean free path for photons. As a result, the initial temperature distribution within the specimen decays exponentially. The penetration effect and the resulting decaying temperature distribution are taken into account in the porous materials model”.

Please add more description of Figure 2.

The description of the DSC traces in Figure 2 has been reported in the text as follow: The PLA filament showed a cold crystallization at 107.4 °C, followed by a melting peak at 171.4 °C. Differently than the melting, that seems not influenced by the MCFs content, the cold crystallization temperature of the composites decreases with the inclusion of MCFs, as can be easily seen from Figure 2 and from the data shown in Table 2. From the same Figure 2, it is worth to note a slight increase in the glass transition of the filaments with the increase of MCFs.

Please revise the logic of the writing order of the materials and methods, results, and discussion parts, referring to other papers in this journal. May some sentences in results need to be put in another part.

Probably the reviewer is right, but the text has been structured following the Journal template.

Discussion:

The discussion has been modified following the reviewer’s suggestion

 Conclusion

Please simplify this part.

It is difficult to meet this suggestion, the conclusion is just very synthesized. We agree with the reviewer that after the discussion, probably this part could be not necessary but we followed the suggestion reported in the Journal template: “This section is not mandatory but can be added to the manuscript if the discussion is unusually long or complex”

Round 2

Reviewer 1 Report

Authors have answered my comments satisfactorily, and the manuscript is now publishable in Entropy.

Reviewer 2 Report

The revised manuscript basically meets the requirements and can be published.